# Ethical challenges in palliative sedation of adults: protocol for a systematic review of current clinical practice guidelines

Martyna Tomczyk  ,[1] Cécile Jaques,[2] Ralf J Jox[1,3]

¹Institute of Humanities in Medicine, Lausanne University Hospital and University of Lausanne, Lausanne, Switzerland
²Medical Library, Lausanne University Hospital and University of Lausanne, Lausanne, Switzerland
³Palliative and Supportive Care Service, Chair in Geriatric Palliative Care, Lausanne University Hospital and University of Lausanne, Lausanne, Switzerland

**Correspondence to**
Dr Martyna Tomczyk;
Martyna.Tomczyk@chuv.ch

## ABSTRACT

**Introduction** This study aims to identify the full spectrum of ethical challenges of all forms of palliative sedation for adults as presented in current clinical practice guidelines (CPGs) and to determine whether CPGs specify ethical challenges of this therapy for patients with cancer and non-cancer and, if so, how exactly they do this. To the best of our knowledge, no studies have yet investigated this topic. The purpose is purely descriptive; our aim is not to make any kind of normative judgements on these challenges. Nor is our aim to assess the quality of the CPGs.

**Methods and analysis** We will perform a systematic review of CPGs on palliative sedation for adults via five electronic databases, grey literature search tools, citation tracking and contact with palliative care experts. Current CPGs accredited by an international, national or regional authority, published in English, German, French, Italian or Polish, from 2000 to the date of the search, will be subjected to content analysis at the textual, linguistic and thematic levels.

**Ethics and dissemination** This is a protocol for a systematic review and no human will be involved in this research. Therefore, ethics approval and consent to participate are not applicable to this context. This study protocol is reported in accordance with the Preferred Reporting Items for Systematic Review and Meta-Analysis for Protocols criteria and registered on PROSPERO. Moreover, the integral version of this study protocol is published as a preprint on Research Square. The results of this study will be actively disseminated through peer-reviewed journals and books, international, national and local conference presentations, social media and media in general.

## STRENGTHS AND LIMITATIONS OF THIS STUDY

⇒ Clinical practice guidelines (CPGs) will be identified *via* multiple research strategies, including direct contact with palliative care experts across the word and publication of the research call on the websites and in the newsletters of palliative care associations.
⇒ CPGs previously identified on the internet will be sent to palliative care experts for confirmation.
⇒ In-depth textual, thematic and linguistic analysis of the CPGs will be performed. Only CPGs published in one of the five languages that we know fluently will be included in the analysis, most national CPGs being published in the official language of the country.
⇒ The main methodological limitation is linked not to our methodology but to the specificity of the CPGs that relate to palliative sedation without conceptualising this practice in the same way.

## INTRODUCTION

In the context of adult palliative care, sedation is the monitored use of medication intended to induce a state of decreased or absent awareness, in order to relieve the suffering of patients with cancer and non-cancer with otherwise refractory distress.[1 2] This therapy is used in various settings and, consequently, differs in clinical modalities, such as sedating drugs, depth and duration, indications and concomitant withdrawal or withholding of artificial nutrition and hydration. Sedation is usually qualified as light, mild or deep, and brief, intermittent or continuous until death. It can also be qualified as being proportionate or precipitous.[3] Conceptually, 'palliative sedation' is often used as a general term that encompasses all forms of sedation in the population of patients under palliative care.[1] Consistent terminology and definitions are lacking to date, which is a source of much ambiguity, confusion and controversy in clinical practice and research.[4 5] For example, it is difficult to know the exact prevalence of palliative sedation as data vary widely, from 1% to 88%, in the palliative care literature.[6 7]

Clinically, sedation is an important cornerstone of palliative therapy. However, it remains one of the most debated medical practices in the context of palliative care, it is highly complex and requires many multiprofessional discussions, prudent application, broad clinical experience and good practice.[1]

In the last three decades, many clinical practice guidelines (CPGs) and position

BMJ

statements have been developed by international medical associations,[1 8] national[9] or regional scientific societies[10] and local institutions.[11] These guidelines are developed not only to help palliative care physicians address the challenges related to this practice but also to close the gap between research and practice and, *in fine*, to improve care for patients and their relatives.[12]

Systematic reviews of the guidance documents have been performed[13–16] and relevant texts written in English,[14] English and German,[15] English, Dutch and Italian[13] and French[16] have been included in the analysis and their methodological quality assessed.[13–15] These reviews essentially focus on the many clinical issues of palliative sedation, such as indications, choice of medication and dosage, continuation of life-prolonging therapies, timing/prognosis and level of sedation.[13–15] Although the decision-making process inevitably raises several ethical questions and requires multiprofessional and interdisciplinary discussion, little is known of the ethical issues of each type of sedation. Only one review to date has pursued the aim of exploring controversial ethical aspects of palliative sedation.[15] However, that analysis was limited to patient information and was performed using a general approach, without an in-depth exploration of the challenges involved.

In their review, Gurschick *et al*[14] explicitly excluded documents that predominantly focused on ethical discussions, in order to better explore the clinical practice of palliative sedation. To the best of our knowledge, the full comprehensive spectrum of ethical challenges of all forms of palliative sedation presented in CPGs has not been systematically and transparently explored thus far. Addressing this knowledge gap is important, especially for clinical practice, research and training.

The ethical aspects of palliative sedation have frequently been discussed in the literature since the first such publication in 1990.[17] A systematic literature review of this topic performed in 2010 and updated in 2016 shows four main aspects of palliative sedation as lacking in consensus and debate: consistent terminology, the use of palliative sedation for non-physical suffering, ongoing experience of distress in palliative sedation and the relation between palliative sedation, euthanasia and the hastening of death.[18] However, the ethical challenges of each form of palliative sedation have not been explored in depth. A recent systematic review that is focused on continuous sedation until death explores only the quality improvement initiatives for this practice.[19]

As previously noted, there are various forms of palliative sedation. Continuous deep sedation until death (CDSUD), considered 'an extreme facet of end-of-life sedation',[20] is the most controversial, at both the clinical and ethical levels, and is widely explored and debated in the literature.[20] In general, despite some conceptual differences, associations have been drawn between CDSUD and assisted dying that hastens death, such as euthanasia or assisted suicide. However, CDSUD as a medical therapy raises several other important clinical and ethical questions. For example, it is not always specified whether this form of sedation is indicated for psychoexistential distress and, if so, whether it adequately relieves this type of suffering.[21]

Other types of palliative sedation, such as temporary or intermittent sedation, although rarely discussed in the literature, also raise ethical questions. For example, it is not clear how to inform the patient and her/his family that lucidity may not be restored, that symptoms may reoccur, or that death may intervene during a type of sedation intended as temporary.

According to the WHO, palliative care, although initially limited to patients with cancer, is intended for all patients with incurable and/or life-threatening diseases, whether cancer or not.[22] However, several studies show that, in practice, access to palliative care services, and, thus, effective symptom control, is still more difficult for patients with non-cancer.[23] Consequently, research on palliative sedation is essentially based on studies concerning patients with cancer. Likewise, there are only specific CPGs on palliative sedation for patients with cancer.[8] Nevertheless, one study focuses on the practice of continuous palliative sedation for both patients with cancer and non-cancer and suggests several differences in this practice for the two populations.[2] Thus, it is possible that there are also variations in the ethical issues regarding this therapy. To the best of our knowledge, no systematic review of CPGs on palliative sedation has, thus, far explored the ethical issues of this therapy for patients with cancer and non-cancer alike.

Palliative sedation and, more broadly, palliative care, is influenced by the culture of the country and region in which it is practised, not only by the legal and social contexts but also by the patient's culture and that of the interdisciplinary and multiprofessional team involved in her/his care.[14] Thus, the ethical issues of palliative sedation are likely to be in line with the wider context. For example, the legal regulations on CDSUD in France have led palliative care experts to elaborate specific guidelines on this practice.[24]

It is possible that cultural elements are noted in national CPGs and applicable to the country and are not, consequently, directly translatable from one country to another. However, it is interesting to know and compare these questions in different countries, not only English-speaking regions, in order to enrich and develop international reflection in this field.

The aim of our review is to identify systematically, transparently and comprehensively the full spectrum of ethical challenges of all forms of palliative sedation for adults as presented in CPGs. This study also aims to determine whether CPGs explicitly specify the ethical challenges of this therapy for patients with cancer and non-cancer and, if so, exactly how they do this. To the best of our knowledge, at the time of writing this paper, no studies had yet investigated this topic. The purpose is purely descriptive; our aim is not to make any kind of normative judgements on these challenges. Nor is our aim to assess the quality of the CPGs.

In the context of our study, we use the term 'ethical challenges' and elaborate a working definition.

## METHODS AND ANALYSIS
### Study design
We intend to design and perform a systematic review of CPGs on palliative sedation of adults and to focus our analysis on the ethical challenges of all forms of this practice. This systematic review started on 22 June 2021 with the aim of being completed by 31 December 2021. However, for many reasons, especially in respect of the COVID-19 situation, and the deadlines for publication of our call for research recommendations on the websites of palliative care associations, the completion date could not be respected. The planned end date for this systematic review is now 31 May 2022.

### Patient and public involvement
No patients are involved.

### Information sources and search strategy
In the first step, the following five electronic bibliographic databases will be searched: Medline (Ovid), Embase.com, CINAHL with Full Text, APA PsycInfo (Ovid) and Web of Science (All Databases). Strategies will include controlled vocabulary (if available) and free text terms and will be peer reviewed by another librarian using the Peer Review of Electronic Search Strategies checklist[25] (see online supplemental file 1 for the draft Embase strategy). Citations will be integrated in citation management software (Endnote XV.9) for deduplication.

Our systematic CPGs search will be conducted in three main steps by MT, a postdoctoral researcher in the ethics of palliative care, CJ, a medical librarian and RJJ, a palliative care physician and researcher in medical ethics.

The following resources will be used for complementary searches: Trip Database, ECRI Guidelines Trust, Guidelines International Network, NHS Evidence Search, bibnet.org, CisMef, Society guideline links (UpToDate), Google Scholar, Google and the websites of societies of palliative care and medical ethics. If necessary, the corresponding authors will be contacted to obtain more detailed information about guidelines that emerged from these searches.

In complementary resources, a search will be performed in English as well as in French, German, Italian and Polish, according to the list of translated terms (if translatable, translated according to usage). This list is presented in table 1.

In the second step, citation chasing will be carried out on the papers included, in order to identify guidelines that may not have appeared through the database search. Individual and collective members of a palliative care society will then be asked about the guidelines used in their country. Prominent experts in palliative care will be contacted if a country does not have a society for palliative care. Finally, a call will be published on LinkedID (an online professional network).

### Inclusion and exclusion criteria
The inclusion and exclusion criteria are presented in table 2.

### Guidelines selection
In the first step, the titles and abstracts of all CPGs identified will be screened by MT. Relevant CPGs will be

**Table 1** Terms to be used in the Internet search

| English | French | German | Italian | Polish |
|---|---|---|---|---|
| 'Palliative sedation' | 'Sédation palliative' | 'Palliative sedierung' | 'Sedazione palliativa' | 'Sedacja paliatywna' |
| 'Terminal sedation' | 'Sédation terminale' | 'Terminale sedierung' | 'Sedazione terminale' | 'Sedacja terminalna' |
| 'Sedation in palliative care' | 'Sédation en soins palliatifs' | 'Sedierung in der palliative care' | 'Sedazione in cure palliative' | 'Sedacja w opiece paliatywnej' |
| 'Sedation in palliative medicine' | 'Sédation en médecine palliative' | 'Sedierung in der palliativmedizin' | 'Sedazione in medicina palliativa' | 'Sedacja w medycynie paliatywnej' |
| 'Continuous sedation' | 'Sédation continue' | 'Kontinuierliche sedierung' | 'Sedazione continua' | 'Sedacja ciągła' |
| 'Continuous deep sedation until death' | 'Sédation profonde et continue jusqu'au décès' | 'Tiefe kontinuierliche sedierung bis zum tod' | 'Sedazione profonda e continua fino alla morte' | 'Sedacja głęboka ciągła utrzymywana do śmierci' |
| 'Recommendations'/ 'guidelines' | 'Recommandations'/ 'guide' | 'Empfehlungen'/ 'Leitlinien' | 'Linee guida' / 'Raccomandazioni' | 'Rekomendacje'/ 'Wytyczne'/'Zalecenia' |
| 'Practice guidelines' | 'Recommandations / Guide pratique(s)' | 'Praxisempfehlungen'/ 'Praxisleitlinien' | 'Linee guida / Raccomandazioni pratiche' | 'Rekomendacje / Wytyczne / Zalecenia praktyczne' |
| 'Clinical guidelines' | 'Recommandations / Guide clinique(s)' | 'Klinische empfehlungen'/ 'Klinische leitlinien' | 'Linee guida / Raccomandazioni cliniche' | 'Rekomendacje/ Wytyczne / Zalecenia kliniczne' |
| 'Clinical practice guidelines' | 'Recommandations / Guide de pratique clinique' | 'Klinische praxisleitlinien' | 'Linee guida / Raccomandazioni di pratica clinica' | 'Rekomendacje / wytyczne / Zalecenia dotyczące praktyki klinicznej' |

**Table 2** Inclusion and exclusion criteria

| | Inclusion criteria | Exclusion criteria |
|---|---|---|
| Type of document | Explicit statement identifying the document as a 'practice guideline' in line with the definition in MEDLINE. According to this definition, a 'practice guideline' is a 'work consisting of a set of directions or principles to assist the health care practitioner with patient care decisions about appropriate diagnostic, therapeutic, or other clinical procedures for specific clinical circumstances'[30]. | All documents contrary to the definition of a 'practice guideline' published on MEDLINE, such as documents not accredited by a national authority, results of studies, study protocols, systematic literature reviews, descriptions of the procedure, expert opinion without a consensus conference, commentaries, letters and editorials. |
| Subject of document | Palliative sedation for adults (*without* explicitly specifying, such as cancer or geriatric patients), in all contexts of palliative care, such as a palliative care unit, inter-hospital palliative care mobile team, intra-hospital palliative care mobile team or patient's home. | Sedation in other contexts (eg, anaesthesia, intensive medicine, emergency medicine, or radiology) and/or for other populations (eg, neonatology and paediatrics) or explicitly specified (eg, cancer patients). |
| Source | Texts developed by government agencies, associations, organisations, such as professional societies or governing boards, or by the convening of expert panels. | Texts developed by institutions, such as a hospital. |
| Scope | Texts accredited at the international, national or regional level. | Texts not accredited at the international, national, or regional level, such as internal hospital practice guidelines. |
| Target audience | Medical and paramedical staff. | Other than medical and paramedical staff. |
| Language of publication | English, German, French, Italian or Polish (as native languages or used fluently by the authors of this paper). | Published in a language other than English, German, French, Italian, or Polish. |
| Year of publication | From 2000 to the date of the searches. | Before 2000. |
| Version | If there is more than one version of a specific guideline, only the latest and most up-to-date version. If a short and long version exist, only the long version. | All versions not in force currently and/or a short version. |
| Availability | Only full text. | No full text accessible. |
| Definition of sedation | Explicit or implicit definition (two items required: duration and depth of sedation). | Lack of definition, or lack of one or both of the two items required (duration and depth of sedation). |

retrieved and assessed for potential inclusion in accordance with the inclusion and exclusion criteria (1–9) presented in table 2. Where it is unclear how to apply the inclusion/exclusion criteria, discussions between the two researchers involved at this stage (MT and RJJ) will be held and a consensus procedure applied. The corresponding author of the CPGs will be contacted if further information is required.

In the second step, the eligibility of each full text will be assessed and a final decision made regarding whether it will be included in the analysis. This final selection of full texts will be based on all the inclusion and exclusion criteria (1–10) presented in table 2. Potential disagreements will be resolved by discussion between the two researchers.

### Data extraction and analysis

Given the lack of methodological best practice standards for the extraction and analysis of data regarding ethical challenges, original content analysis will be performed in accordance with the objectives of this study by one reviewer (MT) in collaboration with RJJ (for adjudication on unresolved differences). If deemed necessary,

corresponding authors will be contacted to resolve any uncertainties.

In the first step, vertical analysis (text by text) will be carried out in accordance with the analysis grid presented in table 3.

In the second step, transversal analysis of all texts will be undertaken in accordance with the analysis grid presented in table 4.

All analyses will be performed at the textual, linguistic and thematic levels. Two opposite analysis methods - with and without a framework - will be used.

Textual analysis with a pragmatic framework will be performed to identify the formal characteristics of the CPGs and to determine the importance of ethical considerations in these texts (1–2); a range of types of information chosen in advance will be extracted from the CPGs (see below).

Linguistic analysis with a list of the items chosen in advance will be applied to identify all forms of palliative sedation and analyse their definitions.

In contrast, thematic analysis will be performed in two ways. If chapter(s) and/or paragraph(s) explicitly focused

**Table 3** Grid for vertical analysis

| Formal characteristics of the CPGs | |
|---|---|
| Title and subtitle | |
| Scope (international/national/regional) | |
| Source | |
| Language of publication | |
| Year of publication | |
| Number of pages | |
| Number and affiliations of authors | |
| Availability (journal / website) | |
| **Importance of ethical considerations in the CPGs** | |
| Chapter(s) and/or paragraphs *explicitly* focused on ethical considerations | Yes/no |
| Title(s) | |
| Placement(s) in the text | |
| Participation of a society for medical ethics | |
| Participation of ethicists | |
| **If chapter(s) or paragraph(s) explicitly focused on ethical considerations: Spectrum of ethical challenges of all forms of palliative sedation** | |
| Palliative sedation | Items: depth and duration of sedation, target patients, artificial hydration and nutrition, indications.<br>Example:<br>I. 'X': …<br>II. 'XY': …<br>III. 'XYZ': … |
| Ethical challenges | I. Ethical challenges of 'X'<br>*Ethical challenge no. 1/2*:…<br>a. Cancer and/or non-cancer patients:…<br>b. Disciplinary components:…<br>c. Cultural influence:…<br>*Ethical challenge no. 2/2*:…<br>a. Cancer and/or non-cancer patients:…<br>b. Disciplinary components:…<br>c. Cultural influence:… |
| | II. Ethical challenges of 'XY'<br>*Ethical challenge no. 1/1*:…<br>a. Cancer and/or non-cancer patients:…<br>b. Disciplinary components:…<br>c. Cultural influence:… |
| | III. Ethical challenges of 'XYZ'<br>*Ethical challenge no. 1/1*: …<br>a. Cancer and/or non-cancer patients:…<br>b. Disciplinary components:…<br>c. Cultural influence:… |

CPGs, clinical practice guidelines.

on ethical considerations appear in the CPGs, thematic analysis with continuous theming (without a framework) and line-by-line coding of the text will be used to identify possible ethical challenges of palliative sedation in this/ these part(s) of the text. This method will also be used to determine whether CPGs specify ethical challenges in respect of patients with cancer and non-cancer and, if so, how exactly they do this. According to this method, the majority of the themes will not be identified in advance; they will be inductively derived from the texts, without attempting to validate a particular theory or hypothesis.[26] All identified themes will then be grouped and organised into descriptive categories. It is important to be precise in the way the conceptual framework is developed (see below) for thematic analysis and applied to all texts. We chose this method because of the lack of research on this

**Table 4** Grid for transversal analysis

| | CPGs number 1 | CPGs number 2 | CPGs number 3 |
|---|---|---|---|
| **Formal characteristics** | | | |
| Title and subtitle | | | |
| Scope | | | |
| Source | | | |
| Language of publication | | | |
| Year of publication | | | |
| Number of pages | | | |
| Number and affiliations of authors | | | |
| Availability | | | |
| **Importance of ethical considerations** | | | |
| Chapter(s) and/or paragraphs *explicitly* focused on ethical considerations | | | |
| Title(s) | | | |
| Placement(s) in the text | | | |
| Participation of a society for medical ethics | | | |
| Participation of ethicists | | | |
| **If chapter(s) or paragraph(s) explicitly focused on ethical considerations: Spectrum of ethical challenges of all forms of palliative sedation** | | | |
| Palliative sedation | Items: depth and duration of sedation, target patients, artificial hydration and nutrition, indications.<br>Example:<br>I.'X':…<br>II. 'XY':…<br>III. 'XYZ':… | Items: depth and duration of sedation, target patients, artificial hydration and nutrition, indications.<br>Example:<br>I.'X': …<br>II. 'XY':…<br>III. 'XYZ':… | Items: depth and duration of sedation, target patients, artificial hydration and nutrition, indications.<br>Example:<br>I.'X':…<br>II. 'XY': …<br>III. 'XYZ':… |
| Ethical challenges | I.Ethical challenges of 'X'<br>*Ethical challenge no. 1/2*:…<br>a. Cancer and/or non-cancer patients:…<br>b. Disciplinary components:…<br>c. Cultural influence:…<br>*Ethical challenge no. 2/2*:<br>a. Cancer and/or non-cancer patients:…<br>b. Disciplinary components:…<br>c. Cultural influence:… | I.Ethical challenges of 'X'<br>*Ethical challenge no. 1/1*:…<br>a. Cancer and/or non-cancer patients:…<br>b. Disciplinary components:…<br>c. Cultural influence:… | I.Ethical challenges of 'X'<br>*Ethical challenge no. 1/1*:…<br>a. Cancer and/or non-cancer patients:…<br>b. Disciplinary components:…<br>c. Cultural influence:… |
| | II. Ethical challenges of 'XY' and the rest | II. Ethical challenges of 'XY' and the rest | II. Ethical challenges of 'XY' and the rest |
| | III. Ethical challenges of 'XYZ' and the rest | III. Ethical challenges of 'XYZ' and the rest | III. Ethical challenges of 'XYZ' and the rest |

CPGs, clinical practice guidelines.

topic and to enable us to explore our material in depth. Finally, additional thematic analysis of the integral text (and *not only* a chapter(s) and/or paragraph(s) focused on ethical considerations) will be performed to identify ethical challenges chosen by us in advance. Our plan for data extraction and analysis is summarised in figure 1.

## Textual analysis
### Identification of the formal characteristics of the CPGs
The following general characteristics will be searched for and noted: title and subtitle, scope (ie, international,

national or regional), source, language, year of publication, number of pages, number and affiliations of authors and availability (eg, published in a journal or on a website).

### Determination of the importance of ethical considerations in the CPGs
To determine the place of ethical considerations in the texts, chapters and paragraphs that focus *explicitly* on ethical considerations will be sought (eg, a title containing the word 'ethical'/'ethics' or with an obvious

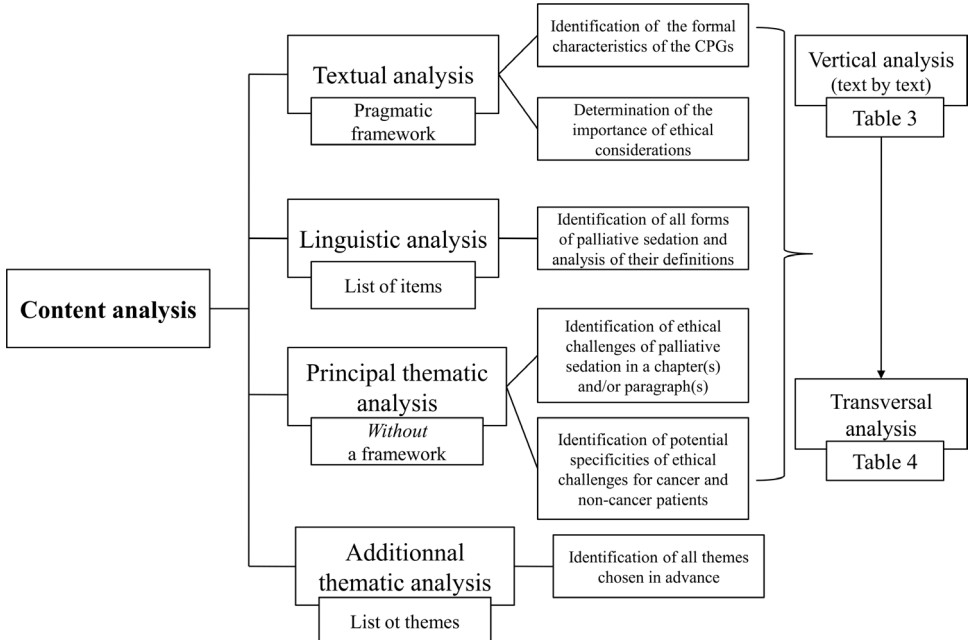

**Figure 1** Data extraction and analysis.

relation to ethics, such as euthanasia). The exact titles and placement in the text will be noted. In addition, the list of authors will be analysed, in order to determine whether and what kind of ethics expertise has been integrated into the CPGs (eg, the participation of a society for medical ethics or ethicists in the development of the CPGs).

### Linguistic analysis

Our linguistic analysis will aim to identify diverse forms of palliative sedation and to explore the terminology and definitions conceptually. In order to incorporate as many types of palliative sedation as possible, a working definition of this practice will not be employed. Initially, a linguistic analysis will be performed to identify all forms of palliative sedation presented in the CPGs. Their definitions, if available in the text, will be analysed according to the several items issued in a previous linguistic study[5]: depth and duration of sedation, sedating drugs, target patients, withdrawal or withholding of artificial nutrition and hydration and indications. In the next step, ethical challenges of each type of sedation identified will be searched for in the full text.

### Thematic analysis
#### Principal thematic analysis

Our principal thematic analysis will aim to identify the full spectrum of possible ethical challenges of all forms of palliative sedation in a chapter(s) and/or paragraph(s) focused on ethical considerations. The analysis will also aim to determine whether the CPGs identified specify the ethical challenges of this therapy in respect of patients with cancer and non-cancer and, if so, how exactly they do this.

### Conceptual framework

Medical ethics can be defined in several ways. In the specialist literature, several terms are used to refer to ethical questions, such as 'ethical issues', 'ethical aspects', 'ethical challenges', 'ethical dilemmas', 'ethical considerations', 'ethical reflection' and 'ethical risks'; often without explicit definition. Thus, in research ethics, the most basic (and paradoxical) question would always be 'which is an ethical question, which is not?'. We understand 'ethics' as rational reflection, both individually and collectively, on moral issues. Moral issues are those that refer to the norms and values that apply to human beings as human beings.[27]

For the purpose of our study, the term 'ethical challenges', which is borrowed from Kahrass et al,[28] is chosen and its working definition is elaborated by our research team (MT, CJ and RJJ). We did not adopt the definition proposed by Kahrass et al[28] because it has a limited focus on principlism. This ethical approach has been elaborated by Beauchamp and Childress[29] and is based on four general principles: beneficence, non-maleficence, respect for autonomy and justice. Currently, this approach is largely and explicitly used in theoretical academic literature and in clinical ethics. However, these principles always have to be adapted to the specific medical and personal context of the individual patient. According to the definition in MEDLINE, CPGs are 'work[s] consisting of a set of directions or principles to assist the healthcare practitioner with patient care decisions about appropriate diagnostic, therapeutic or other clinical procedures for specific clinical circumstances'.[30] Thus, it is not certain that the four ethical principles will be explicitly presented in CPGs. If they are *implicitly* presented in CPGs, their identification and qualitative analysis may be very imprecise,

subjective and even erroneous. In addition, the systematic review performed by Schofield et al,[31] which focuses on the ethical challenges reported by specialist palliative care practitioners in their clinical practice, shows that practitioners use different approaches for ethical reflections, not only the 'four principles' of Beauchamp and Childress.[29]

In our study, an 'ethical challenge' is defined as a relevant difficult situation (purely clinical or not) with regards to palliative sedation that provokes a question, an uncertainty, a controversy or a risk that ethical norms will be transgressed. This situation may also require making a choice between two or more possibilities and needs great mental effort in order to be conducted successfully. According to this working definition, the following questions can be considered an 'ethical challenge': 'Should we accede to a patient's request to induce CDSUD for existential distress?' and 'Should we induce palliative sedation in response to requests for assisted dying, such as euthanasia or assisted suicide?'.

### Analysis

In accordance with the conceptual framework, ethical challenges will be considered using an interdisciplinary approach, but the disciplinary components of each challenge (eg, medical, moral, legal and communicational) will be identified and analysed. Potential cultural influences, such as the legal and social contexts of a country, will also be explored. The ethical challenges of palliative sedation for patients with cancer and/or non-cancer will then be searched for. If they are identified for both populations, their potential similarities and differences will be analysed.

#### Additional thematic analysis

Our additional thematic analysis concerns integral texts (and *not only* chapter(s) and/or paragraph(s) focused on ethical considerations). In each of the CPGs, the following themes will be searched: sedation/euthanasia/assisted suicide, influence of sedation on the duration of a patient's life, sedation for non-physical suffering and information of the patient and family.

### Data synthesis

At the time of writing this study protocol, and to the best of our knowledge, there are no specific methodological standards for the data synthesis of CPGs analysis that is focused on the ethical challenges of a medical procedure. In the literature, authors have proposed a very interesting and useful model for the data synthesis of CPGs analysis focused on ethical issues in line with a disease.[28 32 33] Despite its usefulness, however, we opted not to adopt this approach because it is not fully applicable to a data synthesis that is focused on the ethical challenges of a medical procedure, such as palliative sedation.

Our narrative synthesis will explore the findings within and the relationship between the CPGs included. The ethical challenges of palliative sedation, including their potential comparison in the care of patients with cancer and non-cancer, will be presented according to the types of palliative sedation previously identified in the CPGs. As our systematic review is purely descriptive, we do not intend to carry out theory development.

## DISCUSSION

Our systematic review of CPGs on sedation in adult palliative care aims to explore the full spectrum of possible ethical challenges of all forms of palliative sedation and to determine whether and, if they do, how exactly the CPGs specify the ethical challenges of this therapy for patients with cancer and non-cancer. This review will be of interest to palliative care practitioners of all backgrounds as well as to researchers and educators in palliative care and medical ethics. Our review will provide an initial evidence base for dealing adequately with the ethical issues of this complex but necessary palliative care therapy. To the best of our knowledge, there are no similar systematic reviews available in palliative care. Thus, direct comparison is not possible.

In the literature, especially in publications that are purely theoretical or predominantly conceptual, the ethical issues of palliative sedation are often presented as general reflections, without precision with regards to the clinical characteristics and context of this therapy. However, 'good ethics requires good facts'.[34] For instance, there are several and important differences between temporary and light sedation introduced to relieve physical symptoms, and CDSUD (without proportionality) for existential distress in patients who are not imminently dying. Without a clear distinction, ethical reflection on palliative sedation is immediately biased, even erroneous. We hope that better understanding of the ethical issues of each type of palliative sedation will help reduce this gap and strengthen the methodological rigour of future ethical reflections in this field.

To date, few international guidelines on palliative sedation have been issued.[1 8 14] In contrast, national guidelines have been developed in many countries.[9 10 16 24] In our systematic review, cultural aspects of the ethical challenges of palliative sedation, and the legal and societal contexts of a country in particular, will be identified and analysed. These findings could be useful in enriching debate at the international level and elaborating international guidelines related to this practice. The findings are also intended to be useful for developing international palliative care research involving interventional research between two or more countries. This is important in the context of global development, in which an international approach to palliative care is required to attract greater attention from policymakers.[35] Our findings could also be useful for clinical research at the national level. For instance, it would be interesting to explore whether ethical challenges identified in CPGs are in line with the real-world clinical experience of palliative care practitioners. Previous studies show a gap between the ethical

issues presented in the specialist literature and those reported by palliative care practitioners.[36]

Our findings could also have important implications for education in the ethics of palliative care at all levels. Several studies show the need for training in the ethical aspects of palliative care,[37] so also in palliative sedation. Considered a priority, these courses are often based on a patient's case. However, to support and manage the clinical decision-making process regarding an individual patient, ethical reflection must be engaged at multiple levels, including an international approach. Although not directly applicable for this reflection, the different national influences on the CPGs identified could stimulate this approach positively. It is hoped that our review will further develop the evidence base for curricula in this field.

Our systematic review also aims to determine whether general CPGs specify the ethical challenges of this therapy for patients with cancer and non-cancer and, if so, how exactly they do this. The findings of this part of our study will establish the state of knowledge on this aspect and provide a basis for research and practice. As previously pointed out, there are only specific CPGs on palliative sedation for patients with cancer[8] and the ethical issues, although mentioned in that text, are not deeply developed. Our results could be used in the next update of these CPGs.

Current research on palliative sedation is essentially based on studies involving patients with patients and little is known about palliative sedation, especially the decision-making process, in respect of non-cancer patients, such as patients with dementia. To date, the spectrum of ethical issues in clinical dementia care has been identified and described in the literature[32 38] but has not focused on the specific context of palliative sedation. Our findings could be crucial to developing empirical ethics in this field, in order to generate an understanding of this complex and challenging context.

Finally, our systematic review could bring to the forefront ethical questions from clinicians and practitioners that have not yet been addressed in the ethics literature (eg, when and in what way to inform patients and relatives of the option of palliative sedation). Inversely, CPGs might lack important ethical points that have been extensively discussed in the ethics literature, which would show a lack of transfer from ethics to practice. It would, thus, highlight points that need to be transferred in both directions.

In conclusion, our study will add several new pieces of information to the discussion of the ethical challenges of palliative sedation and, consequently, how current debates on this practice should be approached and addressed.

## ETHICS AND DISSEMINATION

This is a protocol for a systematic review and no human will be involved in this research. Therefore, ethics approval and consent to participate are not applicable to this context.

This systematic review protocol is reported in accordance with the Preferred Reporting Items for Systematic Reviews and Meta-Analyses for Protocols (PRISMA-P)[39] (see online supplemental file 2 for the checklist) and is registered on the International Prospective Register of Systematic Reviews (PROSPERO) as of 22 June 2021 (registration number: CRD42021262571).[40] Moreover, the integral version of our protocol was published as a preprint on Research Square as of 15 December 2021.[41] The review will be reported in line with the PRISMA statement.[42]

The results of this systematic review will be disseminated through various forms of communication strategy, in order to be easily accessible in several contexts at the international, national and local levels and to a diverse range of stakeholders. For example, this will essentially be an academic context (ie, peer-reviewed journal articles and books in English and in the other languages that we know as well as conference and workshop presentations) but also social media (ie, LinkedIn, Facebook and Twitter), general or medical media (ie, newspapers, radio and television) and the internet (ie, the websites of the Institute of Humanities in Medicine of the Lausanne University Hospital in Switzerland and of the Pallium Foundation, located in the canton of Vaud in Switzerland). Finally, our public profile on PROSPERO will be updated and the main results will be published.

**Acknowledgements** The authors would like to express their gratitude to Liz Eggleston, native English speaker and professional proofreader, for helping. In addition, the first author thanks Prof. Jacek Łuczak (1934–2019) from the Palium Hospice and Chair and Department of Palliative Medicine at the Poznań University of Medical Sciences in Poland for his invaluable reflections on palliative sedation. Finally, the authors would also like to give their warm and sincere thanks to the Editor and the two reviewers for their expertise and time, which have helped improve the quality of the manuscript.

**Contributors** MT and RJJ conceived and designed the study; they are the guarantors. MT, CJ and RJJ contributed substantially to the development of the methodological section and to the discussion. MT wrote the manuscript with input from both the other co-authors (CJ and RJJ) MT, CJ and RJJ read, provided feedback and approved the final version of this manuscript.

**Funding** This systematic review is funded by a grant (grant number: not applicable) from the Pallium Foundation (Canton of Vaud, Switzerland). The research is independent of any involvement from this sponsor.

**Competing interests** None declared.

**Patient and public involvement** Patients and/or the public were not involved in the design, or conduct, or reporting, or dissemination plans of this research.

**Patient consent for publication** Not applicable.

**Provenance and peer review** Not commissioned; externally peer reviewed.

**ORCID iD**
Martyna Tomczyk http://orcid.org/0000-0002-5824-2411

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
