## [Reviewer comments · BMJ Open]

ARTICLE DETAILS

TITLE (PROVISIONAL)	Ethical challenges in palliative sedation of adults: protocol for a systematic review of current clinical practice guidelines
AUTHORS	Tomczyk, Martyna; Jaques, Cécile; Jox, Ralf

VERSION 1 – REVIEW

REVIEWER	Morita, Tatsuya Seirei Mikatahara Hospital, Palliative and Supportive Care
REVIEW RETURNED	26-Dec-2021

GENERAL COMMENTS	The topic of this planned systematic review is of great interest, and the protocol paper is clear and well-written. Only minority concern is that language is limited to some language, but this is an acceptable limitation. This reviewer has no specific comments.
---

REVIEWER	Mitra, Sukanya Govt Med Coll
REVIEW RETURNED	28-Jan-2022

GENERAL COMMENTS	Overall a well written protocol, carefully planned and designed, and searching for relevant answers to an important question. Just a few minor clarifications should make the paper clearer to the reader and easier to capture. 1. Please explain why this is a systematic review and not a narrative or scoping review. This is especially because, as the authors write, there will be no quality assessment of the source documents, which is usually an important element of a systematic review.2. In the Abstract and Table 2 point 4 (scope), the word "validated" has been used, e.g., "CPGs validated by an international, national, or regional authority." The use of the word "validated" in this context might be confusing for some readers because the word validation may refer to other constructs not usually applied to CPGs (e.g., validation of a diagnosis, instrument or test etc.). Probably what the authors mean is these CPGs should be "accredited" or "ratified" or "endorsed" by the relevant authorities.3. In Table 2, point 2 (subject of document), it is not clear why the authors would include only those palliative sedation contexts "without explicitly specifying, such as cancer or geriatric patients". Kindly justify in the table or in text.4. It would be helpful to provide a few more examples of what the authors call "ethical challenges", because the entire work rests on identifying these challenges. The two examples provided (pages 17 and 18) both refer to extremes of palliative sedation (CDSUD, euthanasia). The challenges would become more complex when considering, for example, intermittent and/or mild sedation without
---

	the goal of CDSUD or assisted dying in any form but simply for, say, temporary distress relief. Similarly, example of socio-cultural aspects of defining an ethical challenge can be very helpful. The authors may consider adding a box/panel to illustrate several exemplars of ethical challenges likely to be identified in their research.
--	--

VERSION 1 – AUTHOR RESPONSE

Reviewer: 1

The topic of this planned systematic review is of great interest, and the protocol paper is clear and well-written.

Thank you very much for your nice comments. This feedback is very gratifying.

Only minority concern is that language is limited to some language, but this is an acceptable limitation.

We completely agree with you. We decided to include in our systematic review only the guidelines published in English, German, French, Italian or Polish. These are the languages that we know fluently and we believe that a good knowledge of a language is essential to perform an in-depth linguistic and thematic analysis at the explicit and implicit levels. Initially, we planned to include recommendations published in other languages (Spanish, Hebrew, etc.) and have them translated by a professional translator. After reflection, we abandoned this complex idea.

Reviewer: 2

Overall a well written protocol, carefully planned and designed, and searching for relevant answers to an important question. Just a few minor clarifications should make the paper clearer to the reader and easier to capture.

Thank you very much for such a positive comment. Of course, we will aim to clarify our manuscript.

Please explain why this is a systematic review and not a narrative or scoping review. This is especially because, as the authors write, there will be no quality assessment of the source documents, which is usually an important element of a systematic review.

Thank you very much for drawing our attention to this important methodological aspect of our protocol and for your relevant question. Indeed, before developing this research protocol, we frequently discussed the relevance of the assessment of the quality of CPGs, and the relevance of this methodological aspect in the context of the identification and analysis of the possible ethical challenges of palliative sedation in these texts. We hesitated between using a systematic review and a narrative or scoping review. Finally, we favoured a systematic review because of the research question and purpose and the type of documents examined.

Our research is only focused on one type of the text – CPGs. The objective of the research was to *identify* CPGs for palliative sedation across the world and to *synthesize and summarize* the existing ethical challenges of palliative sedation. So, we chose a structured research process and the rigorous and transparent methods of a systematic review. We wanted to be as precise, transparent, and comprehensive as possible.

According to Munn et al.¹, the indications for scoping reviews are as follows:

- As a precursor to a systematic review.
- To identify the types of evidence available in a given field.
- To identify and analyse knowledge gaps.
- To clarify key concepts/definitions in the literature.
- To examine how research is conducted on a certain topic or in a particular field.
- To identify key characteristics or factors related to a concept.

These indications are very interesting but not applicable to our research.

We decided not to assess the quality of the documents because our research question concerns *only ethical* challenges (identification, importance, characteristics, nature, etc.) and *not clinical* evidence, such as the dose of medication or the effectiveness of a practice.

In the Abstract and Table 2 point 4 (scope), the word "validated" has been used, e.g., "CPGs validated by an international, national, or regional authority." The use of the word "validated" in this context might be confusing for some readers because the word validation may refer to other constructs not usually applied to CPGs (e.g., validation of a diagnosis, instrument or test etc.). Probably what the authors mean is these CPGs should be "accredited" or "ratified" or "endorsed" by the relevant authorities.

Many thanks for your helpful comment. We fully agree with you; CPGs should be 'accredited'/'ratified'/'endorsed' by the relevant authorities. We have replaced 'validated' with 'accredited'.

In Table 2, point 2 (subject of document), it is not clear why the authors would include only those palliative sedation contexts "without explicitly specifying, such as cancer or geriatric patients". Kindly justify in the table or in text.

Thank you for this important comment. We decided to include only the CPGs on palliative sedation in general, 'without explicitly specifying, such as cancer or geriatric patients', for two reasons. Firstly, our systematic review aims to determine whether CPGs explicitly specify the ethical challenges of this therapy for cancer and non-cancer patients and, if so, exactly how they do this. If we include specific CPGs on palliative sedation for cancer patients, we cannot reach this objective. Secondly, a particular context, such as geriatric or neurological patients, can potentially raise specific ethical challenges. So, it is important that the thematic scope be more or less homogeneous, and without specific contexts.

It would be helpful to provide a few more examples of what the authors call "ethical challenges", because the entire work rests on identifying these challenges. The two examples provided (pages 17 and 18) both refer to extremes of palliative sedation (CDSUD, euthanasia). The challenges would become more complex when considering, for example, intermittent and/or mild sedation without the goal of CDSUD or assisted dying in any form but simply for, say, temporary distress relief. Similarly, example of socio-cultural aspects of defining an ethical challenge can be very helpful. The authors may consider adding a box/panel to illustrate several exemplars of ethical challenges likely to be identified in their research.

¹ Munn, Z., Peters, M.D.J., Stern, C. *et al.* Systematic review or scoping review? Guidance for authors when choosing between a systematic or scoping review approach. *BMC Med Res Methodol* 18, 143 (2018). <https://doi.org/10.1186/s12874-018-0611-x>

We are very grateful to you for this suggestion and we completely agree with you. We explain this and add examples in the main text, in the 'Data analysis' section. We have also modified our 'Grids for analysis'.